# Anti-Inflammatory Activity of Soluble Epoxide Hydrolase Inhibitors Based on Selenoureas Bearing an Adamantane Moiety

**DOI:** 10.3390/ijms231810710

**Published:** 2022-09-14

**Authors:** Vladimir Burmistrov, Christophe Morisseau, Denis A. Babkov, Tatiana Golubeva, Dmitry Pitushkin, Elena V. Sokolova, Vladimir Vasipov, Yaroslav Kuznetsov, Sergey V. Bazhenov, Uliana S. Novoyatlova, Nikolay A. Bondarev, Ilya V. Manukhov, Victoria Osipova, Nadezhda Berberova, Alexander A. Spasov, Gennady M. Butov, Bruce D. Hammock

**Affiliations:** 1Department of Entomology and Nematology, Comprehensive Cancer Center, University of California, Davis, CA 95616, USA; 2Department of Chemistry, Technology and Equipment of Chemical Industry, Volzhsky Polytechnic Institute (Branch), Volgograd State Technical University, 404121 Volzhsky, Russia; 3Department of Pharmacology & Bioinformatics, Scientific Center for Innovative Drugs, Volgograd State Medical University, 400131 Volgograd, Russia; 4Institute of Cytology and Genetics SB RAS, 630090 Novosibirsk, Russia; 5Specialized Educational Scientific Center, Novosibirsk State University, 630090 Novosibirsk, Russia; 6N.I. Vavilov All-Russian Institute of Plant Genetic Resources, 42, 44, Bolshaya Morskaya Street, 190000 St. Petersburg, Russia; 7Research Center for Molecular Mechanisms of Aging and Age-Related Diseases, Moscow Institute of Physics and Technology, Institutsky Lane 9, 141700 Dolgoprudny, Moscow Region, Russia; 8Toxicology Research Group of Southern Scientific Centre of Russian Academy of Science, 41 Chekhova Str., 344006 Rostov-on-Don, Russia; 9Department of Chemistry, Astrakhan State Technical University, 16 Tatisheva Street, 414056 Astrakhan, Russia

**Keywords:** soluble epoxide hydrolase, inhibitor, adamantane, selenourea, anti-inflammatory

## Abstract

The inhibitory potency of the series of inhibitors of the soluble epoxide hydrolase (sEH) based on the selenourea moiety and containing adamantane and aromatic lipophilic groups ranges from 34.3 nM to 1.2 μM. The most active compound **5d** possesses aliphatic spacers between the selenourea group and lipophilic fragments. Synthesized compounds were tested against the LPS-induced activation of primary murine macrophages. The most prominent anti-inflammatory activity, defined as a suppression of nitric oxide synthesis by LPS-stimulated macrophages, was demonstrated for compounds **4a** and **5b**. The cytotoxicity of the obtained substances was studied using human neuroblastoma and fibroblast cell cultures. Using these cell assays, the cytotoxic concentration for **4a** was 4.7–18.4 times higher than the effective anti-inflammatory concentration. The genotoxicity and the ability to induce oxidative stress was studied using bacterial *lux*-biosensors. Substance **4a** does not exhibit genotoxic properties, but it can cause oxidative stress at concentrations above 50 µM. Put together, the data showed the efficacy and safety of compound **4a**.

## 1. Introduction

The human soluble epoxide hydrolase (sEH) is an enzyme widely distributed in mammalian tissues and involved in the metabolism of natural epoxy fatty acids, including arachidonic acid epoxides formed through oxidation by CYP enzymes [1]. These endogenous epoxides of arachidonic acid have multiple beneficial physiological activities [2]. The sEH converts epoxides via the addition of a water molecule into the corresponding proinflammatory vicinal diols. Thus, inhibition of the sEH enzyme could be beneficial in the treatment of numerous cardiovascular (e.g., hypertension and heart attack), neuronal (e.g., acute and chronic pain), pulmonary (e.g., asthma) and renal diseases (e.g., kidney failure) [3,4].

Among the thousands of various sEH inhibitors (sEHI), ureas and amides make up the majority [5,6,7]. A typical sEH inhibitor consists of a primary pharmacophore of the urea or amide type associated with a lipophilic group. Many of these compounds possess high inhibitory activity but, at the same time, are characterized by low water solubility and high melting points, which diminishes their bioavailability and in vivo efficacy. To resolve the existing problems of sEHI, numerous structural modifications were made. Among those modifications were the establishment of the basic structure of inhibitors [8] and various alterations of the lipophilic part of its molecules [7,9,10]. Compound EC5026 (Figure 1) containing the 3-fluoro-4-trifluorometoxyphenyl lipophilic group is in phase I human clinical trials as a new drug candidate intended to treat neuropathic pain [11]. Additional sEH inhibitors are needed to explore multiple clinical indications that could benefit from the reduction of endoplasmic reticulum stress and resolution of inflammation. Such compounds also facilitate a study of the enzyme catalytic mechanism. The substitution of oxygen in the urea group by other chalcogenes was not systematically investigated.

Recently, we reported the study of thioureas as sEH inhibitors [12]. While being less active (IC_50_ up to 7.2 nM), thioureas are more soluble than ureas, which makes them more bioavailable and thus promising as sEH inhibitors. NMR studies of ureas, thioureas and selenoureas revealed that chemical shifts of selenium-containing compounds are very close to their sulfur analogs [13].

Herein, we studied a wide series of disubstituted selenoureas containing both adamantane and aromatic moieties. The synthesized compounds are characterized by various spacers between the selenourea group and the adamantane fragment. For example, the difference between series **4** and **5** lies in the addition of a methylene spacer between the selenourea group and aromatic fragment for series **5**.

## 2. Results and Discussion

### 2.1. Chemistry

The synthesis of isoselenocyanates was carried out according to a known method from the corresponding amines (**1′** and **2′**) by the action of dichlorocarbene generated from CHCl_3_, followed by the addition of elemental selenium (Figure 1) [14]. The reaction was promoted by phase transfer catalyst Aliquat 336 (Starks’ catalyst) consisting of quaternary ammonium salt N-Methyl-N,N,N-trioctylammonium chloride.

We developed [13] a new method for the isolation and purification of phenyl- (**1**) and benzyl isoselenocyanate (**2**). The effectiveness of the developed method lies in the purification of the target products without the use of a long and laborious process of separation by column chromatography. The process of rapid purification of isoselenocyanates is achieved by filtering the reaction mixture through the thin layer of silica and removing the initial solvent (CH_2_Cl_2_), followed by recrystallization from hexane. The purity was 99+% (GC-MS). The properties of isoselenocyanates **1** and **2** corresponded to the literature data [14].

The synthesized isoselenocyanates **1** and **2** were used to obtain 1-[(adamantan-1-yl)alkyl]-3-phenyl- (**4a–g**) and benzylselenoureas (**5a–g**) by a reaction with adamantyl-containing amines of various structures (**3a–g**, Figure 2 and Figure 3).

The ^1^H NMR spectra of **4a–4g** showed a signal at 7.48–7.96 ppm of the NH proton nearest to the adamantane fragment and a signal at 9.60–10.04 ppm of NH linked to the benzene ring. As for benzyl derivatives **5a–5g**, the latter signal shifts upfield to 7.91–8.45 ppm. The isosteric replacement of the oxygen or sulfur atom in structurally related ureas (thioureas) by selenium is accompanied by the downfield shift of the NH proton signals. It was difficult to record the ^13^C NMR spectrum for reported compounds, namely no C=Se signal was detected. By increasing the scan number from 768 to 7680, we succeeded in detecting a signal at 176.56 ppm. In the spectra of the corresponding urea and thiourea, the C=O and C=S chemical shifts were 153.9 and 178.7 ppm, respectively.

In IR spectra, C–Se stretching vibrations lie in the frequency range 1270–1312 cm^−1^.

### 2.2. Structure-Activity Relationship Study

To our knowledge, no systematic studies of selenoureas as sEH inhibitors have been performed previously. The potency of the compounds was determined against the human recombinant sEH and their solubility assessed in phosphate-buffered saline (Table 1).

Inhibitory activity (IC_50_) of the synthesized selenoureas varies from 34.3 nM to 1.2 µM. The introduction of one methylene group between an adamantyl fragment and a selenourea group leads to a 3.3-fold increase in the inhibitory activity (IC_50_ 435.5 nM for compound **4a** and 131.4 for **4b**, Table 1). Further elongation of this spacer to 1,2-ethylene doubles the potency (IC_50_ 49.1 nM for compound **4d**, Table 1). For the compounds **4a–g**, flexibility at the connection point of an adamantane fragment and a selenourea group tends to be the determining factor affecting the inhibitory potency. For example, compound **4g**, wherein the adamantane fragment connected directly with the urea group via a bridge carbon atom is four times more potent than compound **4a**, in which the connection is made via bridgehead carbon atom. Interestingly, compound **4c** that derived from rimantadine is the least active among the series (IC_50_ 1.2 µM, Table 1), probably due to the shielding of the selenourea group by the 1,1-ethylene spacer. It was assumed that the replacement of the phenyl substituent in series **4** with benzyl will improve the inhibition potency of series **5** due to an increased flexibility, but it turned out to be ineffective. Probably, this flexibility on the other side of the selenourea group leads to less active conformations. While the most potent compound within the studied series, **5d** (IC_50_ 34.3 nM), is bearing a benzyl moiety, **5b**, **5e** and **5g** are less potent than its phenyl-containing analogs. It should be noted that, in most cases, any new methylene spacer between the selenourea group and lipophilic fragments increases the inhibitory activity against sEH. This phenomenon is best observed in a series of selenoureas: **4a**, **5b** and **5d** (Table 1).

Solubility in the sodium phosphate buffer for most of the synthesized compounds lies in the 30–200 μM range. Selenoureas derived from benzyl isoselenocyanate (**2**) show better water solubility then their analogs synthesized from phenylisoselenocyanate (**1**), probably due better flexibility of the molecules.

### 2.3. Tests on Living Cells

Since synthesized compounds were developed with the prospect of their pharmacological use, after in vitro studying the specific activity of soluble epoxide hydrolase inhibition, tests on living cells were carried out. In the case of living cells, both eukaryotic and prokaryotic cells were used.

sEH inhibitors mediate prostaglandins synthesis in immune cells; hence, sEH inhibitors are anticipated to resolve and limit inflammation [16]. For screening designed compounds according to their probable ability to limit inflammation, we used a test on the inhibition of LPS-induced activation of primary murine macrophages. The most prominent anti-inflammatory activity, defined as the suppression of nitric oxide synthesis by LPS-stimulated macrophages, was evident for compounds **4a** and **5b**. At the same time, for **4a**, the cytotoxic concentration was 4.7–18.4 times higher than the effective anti-inflammatory concentration. For compound **5b**, the efficacy and cytotoxicity levels are comparable (Table 2). Other target compounds were found to be essentially inactive as NO synthesis inhibitors. According to the data obtained, there is no direct correlation between the efficiency of the inhibition of sEH and NO synthesis, which requires further studies of the anti-inflammatory activity of the investigated compounds.

As soon as the investigated compounds showed cytotoxicity in the experiment with peritoneal macrophages, their toxic effect on different types of cells was also studied. At the first step, whole-cell bacterial biosensors were used for the assessment of the target compounds’ toxicity against bacterial cells. Both gram-negative and gram-positive bacteria were used. Whole-cell bacterial *lux*-biosensor systems are widespread; they are used to study the general and specific toxicities of various chemical compounds and mixtures, which allows defining the mechanism of toxic action [17,18,19,20,21,22].

**4a** was chosen for these tests as a compound, which showed the significant ability of sEH inhibition in vitro and also anti-inflammatory activity in vivo (NO synthesis inhibition tests) and could be considered as the most prominent for the further use. **4g** was chosen as the isomer of **4a**, which differs only in the way of adamantyl bonding to the selenourea group and is able to inhibit sEH but not NO synthesis. The study was carried out with use of the following whole-cell bacterial biosensors: *Escherichia coli* MG1655 pOxyR-lux, *E. coli* MG1655 pDps, *E. coli* MG1655 pSoxS-lux, *E. coli* MG1655 pColD-lux, *E. coli* MG1655 pAlkA-lux, *E. coli* MG1655 pXen7, *Bacillus subtilis* 168 pNK-AlkA, *B. subtilis* 168 pNK-MrgA and *B. subtilis* 168 pNK-DinC. These biosensors by the quantitation of bacterial stress promoters’ activity, make it possible to determine oxidative stress caused by hydrogen peroxide and the superoxide anion radical, SOS response, DNA alkylation, or general toxicity in Gram-positive and Gram-negative *B. subtilis* and *E. coli* cells. We used concentrations from 1 μM to 1 mM of the studied compounds added to the suspension of biosensor cells. Table 3 shows the maximum induction coefficients—max(K_ind_), calculated according to (2) for the standard toxicants and for the studied compounds.

Table 3 shows that compound **4a** causes only the activation of stress promoter P*_oxyR_*. **4g** does not cause an increase in the luminescence of the used stress-inducible biosensors in concentrations from 1 μM to 1 mM.

Typical kinetic curves for compound **4a** with *E. coli* MG1655 pOxyR cells and **4a** and **4g** with *E. coli* MG1655 pColD cells are shown in Figure 2.

Figure 2A,B show that compound **4a** causes the induction of *E. coli* MG1655 pOxyR biosensor cells when added in concentrations above 100 μM in a dose-dependent manner.

Importantly, all the studied compounds are poorly soluble in water and can form a suspension at 1 mM; moreover, compound **4a** at concentrations above 100 μM presents a substance of dark color and is capable of the partial shielding of bioluminescence. The luminescence decreases the coefficient of the *E. coli* MG1655 pXen7 biosensor with the maximum concentration of compound **4a** at 1 mM to 0.29 ± 0.08, and the other compound does not affect the cell luminescence. This decrease occurs instantaneously and persists over time but does not progress, which illustrates the absence of general toxicity of the studied compounds to bacterial cells (the decrease in the luminescence of bacterial cells, except for the case of shielding, correlates with the viability upon treatment with toxicants).

Thus, the investigated compounds do not show toxicity against bacterial cells and do not activate stress promoters in sub-micromolar concentrations sufficient for EH inhibition.

For compounds with evident anti-inflammatory activity (**4a** and **5b**) and one outgroup compound (**4g**), MTT tests with the human fibroblast cells (MRC-5) and human neuroblastoma cells (SH-SY5Y) were carried out. The studied compounds were added in concentrations of 1, 3, 6, 12, 25, 50, and 100 μM. The obtained results in the form of survival rate graphs with calculated EC_50_ values are shown in Figure 3.

**Figure 3 ijms-23-10710-f003:**
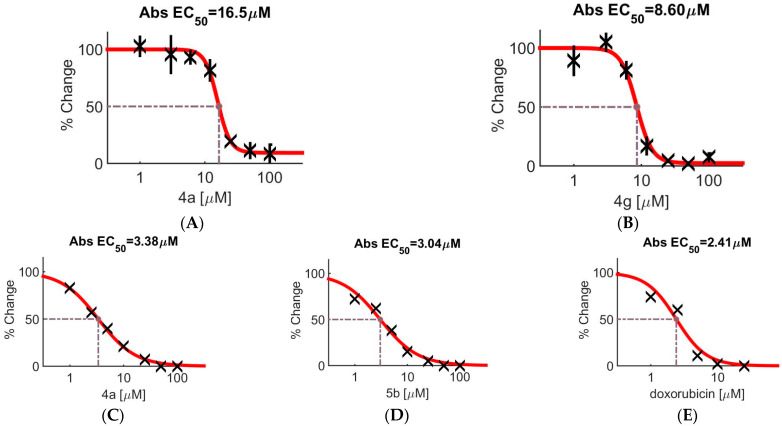
MTT test results for SH-SY5Y cells treated with compounds **4a** (panel **A**) and **4g** (**B**) in different concentrations and for MRC-5 treated with **4a** (**C**), **5a** (**D**), and doxorubicin (**E**) in different concentrations. This figure shows that compounds **4a** and **4g** in micromolar concentrations have a strong toxic effect on the neuroblastoma cell lines. Obtained values are shown by X sign (100%—no toxic effect), standard deviation of obtained value is shown by vertical bar, red line is survival curve fitted by Combenefit software. For compounds **4a** and **4g**, the EC_50_ values were calculated as 16.5 and 8.6 μM (Table 4), respectively. All studied compounds, according to the data from Figure 3, are nontoxic at sub-micromolar concentrations.

**Table 4 ijms-23-10710-t004:** Data for the MTT tests with three cell lines.

Compound	EC_50_ ± SEM, µM		
	MRC-5	SH-SY5Y	C57bl/6j
**4a**	3.38 ± 0.49	16.5 ± 1.5	7.5 ± 1.9
**4g**	–	8.6 ± 0.9	–
**5b**	3.04 ± 0.66	–	17.7 ± 5.66
doxorubicin	2.41 ± 0.15	–	-
Dexamethasone	–	–	>100

Despite the evident general toxicity of the studied compounds against different human cell lines, the experiments with lux-biosensors revealed that the synthesized compounds do not cause toxic effects on bacterial cells in concentrations up to 1 mM. There is no decrease in the luminescence of bacterial cells and no decrease in their growth rate. The absence of toxic effects may be related to the low permeability of the bacterial membrane or cell wall, both Gram-positive and Gram-negative. Oxidative stress was observed only at high concentrations of **4a** (above 100 μM) and was determined by the response of the *E. coli* pOxyR-lux biosensor (Figure 2). The *oxyR* gene promoter is induced by intracellular hydrogen peroxide; the addition of **4a** to the medium leads to the appearance of peroxide inside *E. coli* cells. This can occur both during the oxidation of **4a** with atmospheric oxygen (the effect is typical for some hydrocarbons [23]), resulting in the formation of reactive oxygen species in the environment, or when a compound enters the cell and interacts with various cellular components, such as proteins of the respiratory chain. Compounds do not induce DNA damage that stops the replication fork (SOS response), do not cause the transcriptional activation of DNA glycosylase AlkA (no alkylation), do not cause oxidative damage by superoxide anion radical (i.e., do not activate P*_soxS_*) and do not cause the activation of ferritin gene promoter, which protects DNA from oxidative damage.

For the final safety check of selenoureas, the SOS chromotest was carried out. The standard genotoxic agent 4-nitroquinoline-1-oxide (NQO) was chosen as the positive control. The results are presented as the mean value of induction factor I (c) ± SEM based on the results of three independent experiments, each of which included three replicates (Table 5). It was shown that the studied compounds have no genotoxic activity; therefore, they do not cause DNA damage, leading to a replication block. Thus, the toxic effect of **4a** and **5b** is not determined by genotoxicity.

Many organoselenium compounds exhibit antioxidant activity closely correlated to their ability to scavenge reactive oxygen species (ROS) [24]. It was recently shown that selenoureas of the similar structures possess moderate antioxidant activity [25]. Thus, at the further steps, the complex investigation of the target compounds’ ability to interact with protons, electrons, and radicals was conducted. The ability of compounds **4a–g** and **5a** to transfer a hydrogen atom (HAT) and/or single electron transfer (SET) to DPPH^•^ and ABTS^+•^ with the formation of the more stable radicals was investigated. It was found that compounds **4a–g** and **5a** in reactions with DPPH and ABTS^+•^ radicals exhibit comparable antiradical activity (45–77% inhibition), which is explained by the presence of electron and hydrogen atom transport systems—NH groups and the selenium atom (Table 6).

Despite significant similarities in the structures of the studied compounds, compounds **4e** and **4f** do not exhibit antiradical properties in the reaction with ABTS^+•^. It was previously found that nitrogen-centered sterically hindered DPPH and ABTS radicals prevent reactions from being proceeded both by HAT and SET mechanisms, but to a greater extent, they slow down the process of hydrogen atom transfer [26]. It is possible that derivatives **4e** and **4f** do not exhibit antiradical activity in this test system due to higher dissociation energies of the N-H bond in these compounds.

In addition, it was found that the antiradical activity of the compounds practically does not depend on time, the optical density values sharply decrease during the initial stage of incubation for 10 min (Figure 4), which indicates that the reaction proceeds due to a fast electron transfer (SET mechanism). In this case, compounds **4e** and **4f** are also isolated, which react with the DPPH radical only at the initial moment; a sharp decrease in the absorption intensity is observed within first minute.

To confirm the SET mechanism, the reducing activity of the compounds was studied using tests CUPRAC (Cupric-Reducing Antioxidant Capacity) and FRAP (Ferric ion-Reducing Antioxidant Power). Unlike the DPPH and ABTS tests, these methods are based only on the ability of substances to act as electron donors. It should be noted that the CUPRAC test has several advantages: the manifestation of activity at a pH close to the physiological value and a favorable redox potential; the method is applicable to both lipophilic and hydrophilic antioxidants. The antioxidant potential of the studied derivatives **4a–g** and **5a** is estimated in equivalents of the water-soluble analog of vitamin E—Trolox (TEAC—Trolox Equivalent Antioxidant Capacity), the activity of which is taken as the unit (TEAC_CUPRAC_ = 1, TEAC_FRAP_ = 1).

Compound **4a** in the FRAP test and **4e** in the CUPRAC test demonstrate close to the reference reductivity. Almost all compounds show a reducing ability that exceeds the action of Trolox (TEAC_CUPRAC_ = 1.07–3.39, TEAC_FRAP_ = 1.13–1.81) (Table 6), except for compound **4f**, which exhibits approximately two (FRAP) and five (CUPRAC) times lower activity regarding Trolox. The results obtained are consistent with the data of the DPPH and ABTS tests and indicate the ability of the studied derivatives to act as electron donors, thereby exhibiting antioxidant properties.

One of the most important antioxidant properties of compounds is the ability to chelate transition metal ions, including iron, which participate in the Fenton reaction, forming an active HO^•^ and H_2_O_2_ and contributing to the development of radical processes, respectively, leading to oxidative damage to cells [27]. In this regard, potential antioxidants can act not only as scavengers of active radicals but also have chelating properties. Therefore, the Ferrous Ions Chelating Activity (FIC) of compounds **4a–g** and **5a** were evaluated in this work by their ability to bind iron(II), which is expressed in a decreased amount of the colored ferrozine complex.

In our study, the Fe^2+^-chelating activity of compounds **4d**, **4e** and **5a** was not established; the remaining compounds exhibited chelating activity, which is more than five times less than the activity of the known chelating agent EDTA, taken as 100% (Table 7). The results obtained indicate that these compounds are not capable of inhibiting oxidative processes by binding transition metal ions.

Superoxide radical anion (O_2_^•−^) is the original and one of the most toxic reactive oxygen species, which is formed in a living organism under normal physiological conditions and utilized by the endogenous antioxidant system. O_2_^•−^ performs many functions in the body, but a shift in the balance of antioxidant-active radicals leads to the development of oxidative stress and, as a result, the appearance of various pathological conditions [28]. The ability of derivatives **4a–g** and **5a** to utilize O_2_^•−^ obtained in the xanthine/xanthine oxidase enzymatic system and in the nonenzymatic quinoid oxidation of adrenaline in an alkaline bicarbonate buffer was studied [29].

The superoxide anion radical scavenging activity of derivatives **4a–g** and **5a** was evaluated by the ability to inhibit the reduction of nitro blue tetrazolium (NBT) to blue formazan by trapping O_2_^•−^ generated by xanthine oxidase in the oxidation of hypoxanthine to uric acid. The compounds exhibit insignificant superoxide anion radical scavenging activity; compound **5a** demonstrates the highest activity (>56% inhibition) and the lowest activity is characteristic of compounds **4c**, **4d** and **4e** (Table 7).

The antiradical activity of compounds **4a–g** and **5a** in the reaction of nonenzymatic quinoid oxidation of adrenaline (1-(3,4-dioxyphenyl)-2-methylaminoethanol) in alkaline bicarbonate buffer was evaluated. As a result of this reaction, not only the superoxide anion radical is formed, but also radicals of carbonate–bicarbonate anions, which makes it possible to reveal the total antiradical activity of potential antioxidants [30,31]. The value of adrenaline autoxidation without the addition of compounds **4a–g** and **5a** was taken as 100%, the calculated % inhibition indicating the antiradical activity of the derivatives. The study showed that all compounds exhibited antiradical activity (10–44% inhibition), and compound **5a** showed the highest antiradical activity (Table 7). Thus, we confirmed the antiradical activity of compounds **4a–g** and **5a** with respect to O_2_^–•^ generated in enzymatic and nonenzymatic model systems.

Nitric oxide (NO^•^) is a less active free radical compared to O_2_^–•^, but when NO^•^ interacts with O_2_^–•^, a highly reactive peroxynitrite anion (ONOO-) is formed, which significantly increases the toxicity of these reactive oxygen species [32]. The activity of compounds **4a–g** and **5a** with respect to NO^•^ generated from sodium nitroprusside, which further interacts with dissolved oxygen to form nitrite ions measured using the Griss reagent, was investigated [33]. All compounds exhibit antiradical activity against NO^•^, except for compounds **4a** and **5a** (Table 7). Compound **4c** shows the greatest activity, which is consistent with the results on the antiradical (DPPH^•^ and ABTS^+•^) and reducing activities (CUPRAC and FRAP).

Thus, the antiradical activity of derivatives **4a–g** and **5a** in relation the radicals of DPPH, ABTS^•+^, NO^•^ and O_2_^•−^, iron chelating properties (FIC) and their reducing ability in the CUPRAC and FRAP tests were investigated. The ability of compounds to participate in electron transfer reactions has been established, which is explained by the presence of selenium atoms. Despite the great similarity of the structures of the compounds studied, the greatest activity is characteristic of compound **4c**. The results on the antioxidant activity of this compound indicate the prospects of its use as a potential inhibitor of oxidative processes. The development of new biologically active molecules requires the comprehensive screening of bioactivity and toxicity; therefore, further research is necessary.

## 3. Materials and Methods

### 3.1. Synthetic Procedures [13]

Stock aniline (99.5%, extra pure, CAS 62-53-3), benzylamine (99%, CAS 100-46-9), Aliquat^®^ 336TG (CAS 63393-96-4) and elemental gray selenium (99.5+%, 200 mesh, CAS 7782-49-2) from Acros Organics (Geel, Belgium), and 1-aminoadamantane (97%, CAS 768-94-5), 2-aminoadamantane hydrochloride (99%, CAS 10523-68-9), 1-amino-3, 5-dimethyladamantane hydrochloride (≥98%, CAS 41100-52-1), 1-aminomethyladamantane (98%, CAS 17768-41-1), 1-(adamantan-1-yl)ethan-1-amine (99%, CAS 1501-84-4), triethylamine (BioUltra ≥99.5%, CAS 121-44-8), tryptophan, Mitomycin C, Methyl viologen and methyl methanesulfonate manufactured by Sigma-Aldrich (St. Louis, MO, USA) were used without purification. Hydrogen peroxide was from Iodine Technologies and Marketing (Moscow, Russia). The starting 2-(adamantan-1-yl)ethan-1-amine hydrochloride [34] and 4-adamantylaniline hydrochloride [35] were obtained by known procedures.

The structure of the obtained compounds was confirmed by ^1^H and ^13^C NMR spectroscopy, GC-MS and an elemental analysis. Mass spectra were recorded on a GC-MS Agilent GC5975/MSD 7820 (Agilent Technologies, Santa Clara, CA, USA). ^1^H and ^13^C NMR spectra were taken on a Bruker Avance 600 (Bruker Corporation, Billerica, MA, USA) in DMSO-*d*_6_ solvent; ^1^H chemical shifts are given relative to SiMe_4_. The elemental analysis was performed on Perkin-Elmer Series II 2400 (Perkin-Elmer, Waltham, MA, USA). Melting points were determined on an OptiMelt MPA100 instrument (Stanford Research Systems, Sunnyvale, CA, USA).

**Phenyl isoselenocyanate (1)**. A mixture of 10.23 g (0.11 mol) aniline, 17.46 g (11.8 mL, 0.146 mol) CHCl_3_, 50 mL CH_2_Cl_2_, 4.40 g (0.011 mol) phase transfer catalyst Aliquat 336 and 29.48 g (38.41 mL, 0.737 mol) 50% aqueous NaOH stirred vigorously for 1 h. After the exothermic effect ceased, the mixture was vigorously stirred at reflux for another 8 h. The amine consumption was monitored by GC-MS. After cooling to room temperature, the mixture was washed by 1 N solution of HCl to remove the unreacted aniline. After that, 4.40 g (0.011 mol) of Aliquat 336, 25.08 g (32.67 mL, 0.627 mol) of a 50% NaOH aqueous solution and 10.94 g (0.139 mol) of finely dispersed gray selenium were added to the reaction mass, and the mixture was stirred for 2.5 h at room temperature. The consumption of phenylisonitrile was monitored by GC-MS. Upon completion of the reaction, 30 mL of water and 30 mL of CH_2_Cl_2_ were added to the reaction mixture, unreacted selenium was filtered off and the organic layer was dried over MgSO_4_. After the drying agent was removed, the reaction mass was filtered through a layer of silica gel (1.5 cm) on a glass filter. The solvent was evaporated under reduced pressure, and the desired product was extracted with hexane (50 mL). Yield 7.64 g (42%). ^1^H NMR (DMSO-*d*_6_), δ, ppm: 7.30–7.33 (m, 2H, 2,6-H arom.), 7.34–7.37 (m, 1H, 4-H arom.), 7.38–7.41 (m, 2H, 3,5-H arom.). ^13^C-APT NMR (DMSO-*d*_6_), δ, ppm: 126.12 (2C, 3,5-C arom.), 128.11 (4-C arom.), 129.60 (2C, 2,6-C arom.). MS (EI) m/z: 185 (15.0%, [M + 3]^+^), 184 (10.0%, [M + 2]^+^), 183 (92.0%, [M + 1]^+^), 182 (7.0%, [M]^+^), 181 (45.0%, [M − 1]^+^), 180 (17.0%, [M − 2]^+^), 179 (19.0%, [M − 3]^+^), 103 (35.0%, [Ph-NC]^+^), 77 (100.0%, [Ph]^+^). Anal., %: (C_7_H_5_Se) C 46.20, H 2.80, N 7.65. Calcd., %: C 46.17, H 2.77, N 7.69. M = 182.08.

**Benzyl isoselenocyanate (2)**. Prepared analogously to compound **1** from 11.77 g (0.11 mol) of benzylamine, 17.46 g (11.8 mL, 0.146 mol) of CHCl_3_, 50 mL of CH_2_Cl_2_, 4.40 g (0.011 mol) of phase transfer catalyst Aliquat 336 and 29.48 g (38.41 mL, 0.737 mol) 50% aqueous NaOH. MS (EI) m/z: 199 (24.0%, [M + 3]^+^), 198 (12.0%, [M + 2]^+^), 197 (100.0%, [M + 1]^+^), 196 (9.0%, [M]^+^), 195 (60.0%, [M − 1]^+^), 194 (22.0%, [M − 2]^+^), 193 (26.0%, [M − 3]^+^), 117 (45.0%, [Ph-CH_2_-NC]^+^), 91 (100.0%, [Ph-CH_2_]^+^). Anal., %: (C_8_H_7_Se) C 49.04, H 3.61, N 7.10. Calcd., %: C 49.00, H 3.60, N 7.14. M = 196.11.

**1-(Adamantan-1-yl)-3-phenylselenourea (4a)**. 
To 0.166 g (1.1 mmol) of 1-aminoadamantane (**3a**) in 10 mL of anhydrous 
diethyl ether was added 0.2 g (1.1 mmol) of phenyl isoselenocyanate (**1**) 
and 0.111 g (1.1 mmol, 0.153 mL) of Et_3_N. The reaction mass was 
stirred at room temperature for 8 h. After removing the ether, 10 mL of 1 N HCl 
was added to the reaction mass and stirred for 1 h. After filtration, the 
precipitate was washed with water and dried. Yield 0.168 g (45%), m. p. 202–203 
°C. ^1^H NMR (DMSO-*d*_6_), δ, ppm: 1.61–1.68 (m, 6H, 
Ad), 1.89 (d, 3H, Ad, *J* = 2.7 Hz), 2.08 (d, 3H, Ad, *J* = 14.1 Hz), 
2.29 (s, 3H, Ad), 7.08–7.10 (m, 1H arom.), 7.13–7.17 (m, 1H arom.), 7.31–7.38 
(m, 3H arom.), 7.48 (s, 1H, Ad-NH), 9.60 (s, 1H, NH-Ph). ^13^C-APT 
NMR (DMSO-*d*_6_), δ, ppm: 29.51 (2C, Ad), 29.74 (2C, Ad), 35.69 
(2C, Ad), 36.31 (2C, Ad), 41.31 (2C, Ad), 44.96 (2C, Ad) 123.39 (2C, 2,6-C 
arom.), 129.08 (4-C arom.), 130.11 (2C, 3,5-C arom.) 139.84 (1-C arom.), 176.56 
(C=Se). IR, ν/cm^−1^: 
1288 (C=Se). Anal., %: (C_17_H_22_N_2_Se) C 61.28, H 
6.67, N 8.37. Calcd., %: C 61.26, H 6.65, N 8.40. M = 333.34.

**1-[(Adamantan-1-yl)methyl]-3-phenylselenourea (4b)**. Prepared analogously to compound **4a** from 0.2 g (1.1 mmol) of isoselenocyanate **1**, 0.182 g (1.1 mmol) of 1-aminomethyladamantane (**3b**) and 0.111 g (1.1 mmol) of Et_3_N. Yield 0.223 g (58%), m. p. 197–198 °C. ^1^H NMR (DMSO-*d*_6_), δ, ppm: 1.51 (c, 6H, Ad), 1.61 (d, 3H, Ad, *J* = 12.2 Hz), 1.68 (d, 3H, Ad, *J* = 11.9 Hz), 1.95 (s, 3H, Ad), 3.38 (s, 2H, NH-CH_2_), 7.15 (dt, 1H arom., *J* = 14.7, 8.7 Hz), 7.33 (dt, 2H arom., *J* = 19.8, 7.7 Hz), 7.44 (d, 2H arom., *J* = 7.7 Hz), 7.87 (s, 1H, CH_2_-NH), 9.90 (s, 1H, NH-Ph). IR, ν/cm^−1^: 1276 (C=Se). Anal., %: (C_18_H_24_N_2_Se) C 62.22, H 6.94, N 8.10. Calcd., %: C 62.24, H 6.96, N 8.06. M = 347.36.

**1-[1-(Adamantan-1-yl)ethyl]-3-phenylselenourea (4c)**. Prepared analogously to compound **4a** from 0.12 g (0.66 mmol) of isoselenocyanate **1**, 0.119 g (0.66 mmol) of 1-(adamantan-1-yl)ethan-1-amine (**3c**) and 0.067 g (0.66 mmol) of Et_3_N. Yield 0.101 g (42%). ^1^H NMR (DMSO-*d*_6_), δ, ppm: 1.03 (d, 3H, CH_3_, *J* = 6.8 Hz), 1.56 (c, 6H, Ad), 1.61–1.72 (m, 6H, Ad), 1.98 (br s, 3H, Ad), 4.38 (br s, 1H, CHCH_3_), 7.14–7.18 (m, 1H arom.), 7.32–7.36 (m, 2H arom.), 7.47–7.51 (m, 2H arom.), 7.53 (d, 1H, CH-NH, *J* = 8.6 Hz), 9.71 (s, 1H, NH-Ph). Anal., %: (C_19_H_26_N_2_Se) C 63.11, H 7.22, N 7.78. Calcd., %: C 63.15, H 7.25, N 7.75. M = 361.39.

**1-[2-(Adamantan-1-yl)ethyl]-3-phenylselenourea (4d)**. Prepared analogously to compound **4a** from 0.2 g (1.1 mmol) of isoselenocyanate **1**, 0.237 g (1.1 mmol) of 2-(adamantan-1-yl)ethan-1-amine hydrochloride (**3d**) and 0.111 g (1.1 mmol) of Et_3_N. Yield 0.171 g (43%), m. p. 141–142 °C. ^1^H NMR (DMSO-*d*_6_), δ, ppm: 1.35–1.38 (m, 2H, Ad-CH_2_), 1.53 (c, 6H, Ad), 1.61–1.71 (m, 6H, Ad), 1.94 (br s, 3H, Ad), 3.59 (br s, 2H, NH-CH_2_), 7.16–7.19 (m, 1H arom.), 7.31–7.37 (m, 4H arom.), 7.87 (s, 1H, CH_2_-NH), 9.68 (s, 1H, NH-Ph). IR, ν/cm^−1^: 1282 (C=Se). Anal., %: (C_19_H_26_N_2_Se) C 63.19, H 7.27, N 7.77. Calcd., %: C 63.15, H 7.25, N 7.75. M = 361.39.

**1-[4-(Adamantan-1-yl)phenyl]-3-phenylselenourea (4e)**. Prepared analogously to compound **4a** from 0.2 g (1.1 mmol) of isoselenocyanate **1**, 0.29 g (1.1 mmol) of 4-adamantylaniline hydrochloride (**3e**) and 0.222 g (2.2 mmol) of Et_3_N. Yield 0.107 g (23%), m. p. 163–164 °C. ^1^H NMR (DMSO-*d*_6_), δ, ppm: 1.69–1.76 (m, 6H, Ad), 1.85 (br s, 6H, Ad), 2.05 (s, 3H, Ad), 7.15–7.18 (m, 1H arom.), 7.30–7.33 (m, 4H arom.), 7.34 (d, 5H arom., *J* = 7.3 Hz), 7.40 (d, 2H arom., *J* = 7.3 Hz), 10.04 (s, 2H, 2NH). Anal., %: (C_23_H_26_N_2_Se) C 67.50, H 6.43, N 6.81. Calcd., %: C 67.47, H 6.40, N 6.84. M = 409.44.

**1-(3,5-Dimethyladamantan-1-yl)-3-phenylselenourea (4f)**. Prepared analogously to compound **4a** from 0.2 g (1.1 mmol) of isoselenocyanate **1**, 0.237 g (1.1 mmol) of 1-amino-3,5-dimethyladamantane hydrochloride (**3f**) and 0.222 g (2.2 mmol) of Et_3_N. Yield 0.121 g (30%), m. p. 142–143 °C. ^1^H NMR (DMSO-*d*_6_), δ, ppm: 0.84 (s, 6H, 2CH_3_), 0.99 (t, 4H, Ad, *J* = 7.1 Hz) 1.23–1.38 (m, 4H, Ad), 1.86–2.02 (m, 4H, Ad), 2.15 (s, 1H, Ad), 7.11–7.16 (m, 1H arom.), 7.30–7.40 (m, 4H arom.), 7.79 (s, 1H, NH), 9.91 (s, 1H, NH). Anal., %: (C_19_H_26_N_2_Se) C 63.18, H 7.28, N 7.71. Calcd., %: C 63.15, H 7.25, N 7.75. M = 361.39.

**1-(Adamantan-2-yl)-3-phenylselenourea (4g)**. Prepared analogously to compound **4a** from 0.2 g (1.1 mmol) of isoselenocyanate **1**, 0.206 g (1.1 mmol) of 2-aminoadamantane hydrochloride (**3g**) and 0.222 g (2.2 mmol) of Et_3_N. Yield 0.173 g (47%), m. p. 187–188 °C. ^1^H NMR (DMSO-*d*_6_), δ, ppm: 1.59–1.87 (m, 12H, Ad), 2.04 (s, 2H, Ad), 4.46 (s, 1H, Ad), 7.13–7.19 (m, 1H arom.), 7.31–7.38 (m, 2H arom.), 7.96 (s, 1H, Ad-NH), 9.93 (s, 1H, NH-Ph). IR, ν/cm^−1^: 1300 (C=Se). Anal., %: (C_17_H_22_N_2_Se) C 61.29, H 6.68, N 8.38. Calcd., %: C 61.26, H 6.65, N 8.40. M = 333.34.

**1-(Adamantan-1-yl)-3-benzylselenourea (5a)**. Prepared analogously to compound **4a** from 0.2 g (1.02 mmol) of isoselenocyanate **2**, 0.154 g (1.02 mmol) of 1-aminoadamantane (**3a**) and 0.103 g (1.02 mmol) of Et_3_N. Yield 0.228 g (65%), m. p. 157–158 °C. ^1^H NMR (DMSO-*d*_6_), δ, ppm: 1.63 (q, 6H, Ad, *J* = 12.1 Hz), 2.05 (s, 3H, Ad), 2.18 (s, 6H, Ad), 4.76 (s, 2H, CH_2_-Ph), 7.24–7.27 (m, 1H arom.), 7.28–7.31 (m, 2H arom.), 732–7.36 (m, 2H arom.), 7.43 (s, 1H, Ad-NH), 9.91 (s, 1H, NH-CH_2_). IR, ν/cm^−1^: 1270 (C=Se). Anal., %: (C_18_H_24_N_2_Se) C 62.20, H 6.95, N 8.10. Calcd., %: C 62.24, H 6.96, N 8.06. M = 347.36.

**1-[(Adamantan-1-yl)methyl]-3-benzylselenourea (5b)**. Prepared analogously to compound **4a** from 0.2 g (1.02 mmol) of isoselenocyanate **2**, 0.168 g (1.02 mmol) of 1-aminomethyladamantane (**3b**) and 0.103 g (1.02 mmol) of Et_3_N. Yield 0.242 g (68%), m. p. 158–159 °C. ^1^H NMR (DMSO-*d*_6_), δ, ppm: 1.47 (s, 6H, Ad), 1.53–1.70 (m, 6H, Ad), 1.92 (s, 3H, Ad), 3.38 (s, 2H, CH_2_-Ad), 4.78 (br s, 2H, CH_2_-Ph), 7.24–7.36 (m, 5H arom.), 7.66 (s, 1H, Ad-CH_2_-NH), 8.15 (s, 1H, NH-CH_2_-Ph). Anal., %: (C_19_H_26_N_2_Se) C 63.12, H 7.24, N 7.78. Calcd., %: C 63.15, H 7.25, N 7.75. M = 361.39.

**1-[1-(Adamantan-1-yl)ethyl]-3-benzylselenourea (5c)**. Prepared analogously to compound **4a** from 0.176 g (0.897 mmol) of isoselenocyanate **2**, 0.160 g (0.897 mmol) of 1-(adamantan-1-yl)-ethan-1-amine (**3c**) and 0.091 g (0.9 mmol) of Et_3_N. Yield 0.2 g (59%), m. p. 154–155 °C. ^1^H NMR (DMSO-*d*_6_), δ, ppm: 0.98 (d, 3H, CH_3_, *J* = 6.8 Hz), 1.40–1.70 (m, 12H, Ad), 1.94 (br s, 3H, Ad), 4.27 (br s, 1H, CH-CH_3_), 4.79 (br s, 2H, CH_2_-NH), 7.24–7.28 (m, 1H arom.), 7.30–7.37 (m, 4H arom.), 7.53 (br s, 1H, CH-NH), 8.03 (br s, 1H, CH_2_-NH). Anal., %: (C_20_H_28_N_2_Se) C 64.00, H 7.50, N 7.50. Calcd., %: C 63.99, H 7.52, N 7.46. M = 375.42.

**1-[2-(Adamantan-1-yl)ethyl]-3-benzylselenourea (5d)**. Prepared analogously to compound **4a** from 0.2 g (1.02 mmol) of isoselenocyanate **2**, 0.220 g (1.02 mmol) of 2-(adamantan-1-yl)ethan-1-amine (**3d**) and 0.206 g (2.04 mmol) of Et_3_N. Yield 0.282 g (74%), m. p. 142–143 °C. ^1^H NMR (DMSO-*d*_6_), δ, ppm: 1.30 (br s, 2H, Ad-CH_2_), 1.48 (s, 6H, Ad), 1.56–1.73 (m, 6H, Ad), 1.92 (s, 3H, Ad), 3.55 (s, 2H, CH_2_-CH_2_-NH), 4.83 (br s, 2H, CH_2_-Ph), 7.23–7.27 (m, 1H arom.), 7.28–7.35 (m, 4H arom.), 7.67 (s, 1H, CH_2_-CH_2_-NH), 8.17 (s, 1H, NH-CH_2_-Ph). Anal., %: (C_19_H_26_N_2_Se) C 63.96, H 7.49, N 7.48. Calcd., %: C 63.15, H 7.25, N 7.75. M = 361.39.

**1-[4-(Adamantan-1-yl)phenyl]-3-benzylselenourea (5e)**. Prepared analogously to compound **4a** from 0.2 g (1.02 mmol) of isoselenocyanate **2**, 0.268 g (1.02 mmol) of 4-adamantylaniline hydrochloride (**3e**) and 0.206 g (2.04 mmol) of Et_3_N. Yield 0.323 g (75%), m. p. 147–148 °C. ^1^H NMR (DMSO-*d*_6_), δ, ppm: 1.64–1.79 (m, 6H, Ad), 1.86 (br s, 6H, Ad), 2.05 (s, 3H, Ad), 4.85 (s, 2H, CH_2_-Ph), 7.22–7.27 (m, 3H arom.), 7.31–7.38 (m, 6H arom.), 8.45 (s, 1H, NH-CH_2_-Ph), 9.97 (s, 1H, NH-Ph-Ad). IR, ν/cm^−1^: 1312 (C=Se). Anal., %: (C_24_H_28_N_2_Se) C 68.10, H 6.69, N 6.59. Calcd., %: C 68.07, H 6.67, N 6.62. M = 423.46.

**1-(3,5-Dimethyladamantan-1-yl)-3-benzylselenourea (5f)**. Prepared analogously to compound **4a** from 0.2 g (1.02 mmol) of isoselenocyanate **2**, 0.220 g (1.02 mmol) of 1-amino-3,5-dimethyladamantane hydrochloride (**3f**) and 0.206 g (2.04 mmol) of Et_3_N. Yield 0.092 g (24%), m. p. 112–113 °C. ^1^H NMR (DMSO-*d*_6_), δ, ppm: 0.82 (d, 6H, 2CH_3_, *J* = 11.1 Hz), 1.08–2.12 (m, 13H, Ad), 4.75 (s, 2H, CH_2_-Ph), 7.21–7.35 (m, 5H arom.), 7.56 (s, 1H, NH-Ad), 8.04 (s, 1H, NH-CH_2_). Anal., %: (C_20_H_28_N_2_Se) C 63.96, H 7.49, N 7.50. Calcd., %: C 63.99, H 7.52, N 7.46. M = 375.42.

**1-(Adamantan-2-yl)-3-benzylselenourea (5g)**. Prepared similarly to compound **4a** from 0.2 g (1.02 mmol) of isoselenocyanate **2**, 0.191 g (1.02 mmol) of 2-aminoadamantane hydrochloride (**3g**) and 0.206 g (2.04 mmol) of Et_3_N. Yield 0.242 g (68%), m. p. 139–140 °C. ^1^H NMR (DMSO-*d*_6_), δ, ppm: 1.55–1.61 (m, 2H, Ad), 1.68–1.85 (m, 10H, Ad), 1.97 (br s, 2H, Ad), 4.43 (br s, 1H, Ad), 4.77 (s, 2H, CH_2_-Ph), 7.26–7.30 (m, 1H arom.), 7.32–7.37 (m, 4H arom.), 7.93 (s, 1H, NH-Ad), 8.20 (s, 1H, NH-CH_2_-Ph). IR, ν/cm^−1^: 1276 (C=Se). Anal., %: (C_18_H_24_N_2_Se) C 62.23, H 7.00, N 8.04. Calcd., %: C 62.24, H 6.96, N 8.06. M = 347.36.

### 3.2. Animals

All procedures with animals were carried out under the generally accepted ethical standards for the manipulations of animals adopted by the European Convention for the Protection of Vertebrate Animals used for Experimental and Other Scientific Purposes (1986) and International Recommendations of the European Convention for the Protection of Vertebrate Animals used for Experimental research (1997). All sections of this study adhere to the ARRIVE Guidelines for reporting animal research [36]. Male mice (21–24 g.) aged 4 to 5 weeks were housed 5 per cage in ambient lighting and 60% humidity. Animals had free access to water and food before the study.

### 3.3. Isolation and Treatment of Peritoneal Macrophages

Peritoneal macrophages (PM) were isolated from the peritoneal exudate of 30 male C57bl/6j mice. To accumulate PM, 1 mL of 3% peptone solution was injected intraperitoneally. After 3 days, the mice were euthanized by cervical dislocation. Cells of peritoneal exudate were obtained by aseptic washing of the abdominal cavity with 5 mL of sterile Hanks’s solution (+4–6 °C) without calcium and magnesium ions. The total number and viability of cells were assessed in a Goryaev counting chamber (Russia) with a 0.4% trypan blue staining (Sigma-Aldrich, St. Louis, MO, USA). The cell concentration was adjusted to 1.0 × 10^6^ cells/mL in DMEM (Gibco) supplemented with 2 mM *L*-glutamine (Gibco), 10% heat-inactivated fetal bovine serum (BioClot, Passau, Germany), 100 U/mL penicillin and 100 mg/mL streptomycin (Gibco) and plated 200 μL/well in 96-well transparent plates (SPL Life Sciences Co., Ltd., Pochon, Korea). After 2 h at 37 °C in a humidified atmosphere with 5% CO_2_, wells were washed to remove nonadherent cells. After 24 h of incubation, 20 μL of the supernatants were substituted with 20 μL of solutions of the test compounds, followed with *E. coli* O127:B8 LPS (100 ng/mL final concentration) after 30 min. Experiments were run in 3 independent replicates.

### 3.4. Assay of Nitric Oxide (NO)

The accumulation of nitrite anion (a stable decomposition product of NO produced by iNOS) in the supernatants was determined using a standard Griess reagent [37]. Briefly, 50 μL of supernatants collected 22 h after incubation of PM with the test and control compounds were mixed with 50 μL of 1% sulfonamide in 2.5% H_3_PO_4_ and 50 μL of 0.1% *N*-(1-naphthyl) ethylenediamine in 2.5% H_3_PO_4_. After incubation at 23 °C for 10 min. in an orbital shaker, the optical density was determined at a wavelength of 550 nm with a microplate reader Infinite M200 PRO (Tecan, Austria).

### 3.5. Cytotoxicity Study

The activity of lactate dehydrogenase (LDH) in a cell culture medium served as a marker of membrane permeability and cell death [38]. Aliquots of supernatants were taken after 24 h of inoculation with test compounds, mixed with 250 μL of 0.194 nM NADH solution in 54 mM phosphate-buffered saline (pH 7.5). Then, 25 μL of a 6.48 mM pyruvate solution was added to the mixture. The optical density was followed at a wavelength of 340 nm for 20 min. The MTT test was performed 24 h after the incubation of cells with the tested compounds. Briefly, 20 μL of MTT solution was added to each well and incubated at 37 °C in a humidified atmosphere containing 5% CO_2_ for 4 h. The culture medium was aspirated, the cells were lysed and formazan crystals dissolved in 150 μL of DMSO. The plates were shaken at room temperature for 10 min, and the optical density was measured in a microplate reader Infinite M200 PRO (Tecan Group Ltd., Männedorf, Switzerland) at a wavelength of 565 nm [39].

### 3.6. Investigation of the Cytotoxicity Using MTT Test with Human Fibroblast Cells (MRC-5)

Human fibroblast cells (MRC-5) were used in this study. The cells were cultured in the DMEM medium that contained 10% embryonic calf serum in a CO_2_ incubator at 37 °C. The tested compounds were dissolved in DMSO and added to the cellular culture at the required concentrations. Three wells were used for each concentration. The cells that were incubated without the compounds were used as a negative control, and doxorubicin was used as the positive one. Cells were placed on 96-well microliter plates and cultivated at 37 °C in 5% CO_2_/95% air for 72 h. The cell viability was assessed through an MTT (3-(4,5-dimethylthiazol-2-yl)-2,5-phenyl-2H-tetrazolium bromide) conversion assay [22]. A total of 1% MTT was added to each well. Three hours later, isopropanol was added and mixed for 15 min. Optical density (OD) of the samples was measured on a ThermoFischer multi-well spectrophotometer at a wavelength of 450 nm. The 50% cytotoxic dose (IC_50_) of each compound (i.e., the compound concentration that causes the death of 50% of cells in a culture or decreases the optical density twice as compared to the control wells) was calculated from the data obtained. The results are given as an average value ± a deviation from the average (mean ± standard error of the mean (SEM)). The experimental results are given as average values obtained from three independently conducted experiments.

### 3.7. Bacterial Strains and Plasmids

In this work, biosensor strains based on *E. coli* MG1655 cells and *B. subtilis* 168 cells were used. 

*E. coli* MG1655 cells contain plasmids, where *luxCDABE* genes from *Photorhabdus luminescens* are controlled by stress-inducible promoters:

pColD-lux―P*_colD_* is induced by the SOS response [17]; pDps―P*_dps_* is activated in the presence of DNA-oxidizing agents [40]; pSoxS-lux―P*_SoxS_* is induced in the presence of superoxide anion radicals [41]; pOxyR-lux―P*_oxyR_* is induced in response to the appearance of hydrogen peroxide or organic peroxides [17]; pAlkA-lux―P*_alkA_* is induced in response to DNA alkylation [20,21]. pXen7 contains *P. luminescens* ZM1 *lux* genes controlled by its own promoter, which was used as a constitutive [42].

*B. subtilis* 168 cells contain plasmids with the *luxABCDE* genes from *P. luminescens* with the RBS sequence adapted to Gram-positive bacteria [43,44] under the control of stress-inducible promoters [18]:

pNK-dinC―P*_dinC_* is induced in response to DNA damage resulting in the SOS response; pNK-alkA―P*_alkA_* is activated in response to DNA alkylation; pNK-mrgA―P*_mrgA_* is induced in response to the presence of hydrogen peroxide.

### 3.8. Cultivation Conditions

For the cultivation of *E. coli* MG1655 cells transformed with biosensor plasmids, LB medium was used: 1% bacto-trypton, 0.5% yeast extract, 1% NaCl and 1.5% bacto-agar (to obtain a solid nutrient medium) with the addition of 200 µg/mL ampicillin. For the cultivation of *B. subtilis* 168 cells transformed with biosensor plasmids, BHI medium with tryptophan at a concentration of 50 μg/mL and chloramphenicol at a concentration of 10 μg/mL were used.

Biosensor cells were grown at 37 °C with constant shaking in a shaker incubator at a speed of 200 rpm.

### 3.9. Bioluminescence Measurement

The bioluminescence of cells was measured using a Synergy HT luminometer (Biotek, Winooski, VT, USA) at room temperature. Lux biosensors based on *E. coli* were grown to an early logarithmic phase (OD ≈ 0.1) to achieve a background luminescence approximately 100–1000 RLU at the beginning of the measurements. Lux biosensors based on *B. subtilis* were grown to a logarithmic phase with an optical density OD ≈ 0.6 (and background luminescence of about 10–500 RLU). Biosensor cell cultures were divided into portions of 200 μL in a 96-well plate. Control toxicants and studied compounds were added to the cell suspension at 2 µL per well. Mitomycin C (at final concentration of 30 μM) for biosensors induced by the SOS response (P*_colD_* and P*_dinC_*); methyl methanesulfonate (1 mM) for biosensors induced by DNA alkylation (P*_alkA_*); hydrogen peroxide (100 μM) for biosensors induced by presence of peroxides (P*_oxyR_*, P*_dps_* and P*_mrgA_*) and methyl viologen (100 μM) for P*_soxS_* activation by the presence of superoxide anion radicals. The kinetics of change in luminescence was measured at room temperature.

### 3.10. Calculation of Induction Coefficients for Luminescence of Biosensors

To unify the obtained results and eliminate the differences between the experiments due to different initial luminescence values, the luminescence value in each sample was normalized to the initial value according to Equation (1): (1)$$L_{norm}\left(t,\;x\right)=\frac{L\left(t,\;x\right)}{L\left(0,\;x\right)}$$
where *L_norm_(t,x)*—luminescence normalized to the initial value for the sample *x*, *L(t,x)* is the luminescence of sample *x* at time *t* and *L(*0*,x)* is the initial luminescence of sample *x* (immediately after addition of the studied compound or control toxicant). This normalization considers the optical density of the studied compound and the luminescence shielding caused by it, which appears immediately after the addition and facilitates the analysis of a slower effect associated with the general toxicity and activation of stress-inducible promoters for which the typical time of activation is in a range from several minutes to 3 h.

To determine the luminescence induction coefficient of biosensor cells, observed due to changes in the level of reporter genes transcription, the following equation was used:(2)$$K_{ind}\left(t,\;x\right)=\frac{L_{norm}\left(t,\;x\right)}{L_{norm}\left(t,\;contr\right)}$$
where *K_ind_(t)* is the calculated luminescence induction coefficient of biosensor cells in sample x at time point *t*, and *L_norm_(t,contr)* is the normalized luminescence in the control sample of the same biosensor cells without the addition of the studied compounds.

### 3.11. Evaluation of Toxic Effect of Compounds on Neuroblastoma Cells SH-SY5Y Using the MTT Test

The MTT test to determine the effect of cell viability was performed on the SH-SY5Y neuroblastoma cell line in a 96-well plate.

Cells grown to 80% confluence were trypsinized and plated at a density of 5000 cells per well (100 µL total well volume) and incubated overnight in a CO_2_ incubator. The next day, a 50 mM stock of studied compounds in DMSO was diluted in DMEM and added in the amount of 50 μL per well to obtain final concentrations of 1–100 μM. After 72 h of cell plate incubation, the supernatant was discarded, 30 μL of 4 mM MTT and 50 μL of DMEM were added to each well and the plate was incubated for 2 h. After that, the supernatant was discarded, 100 μL of DMSO was added and, after the dissolution of the formazan crystals, the absorbance at 570 nm was measured on a Synergy HT plate reader (Biotek, Winooski, VT, USA). An empty well with 100 µL DMSO was used for background signal subtraction, and cells in DMEM and cells with 1% DMSO were used as the negative control. Cells with 0.1% hydrogen peroxide was used as the positive control.

### 3.12. Data Analysis

Statistical analysis and graph preparation were performed in Prism 8.0 (GraphPad Software, Inc., San Diego, CA, USA). One-way ANOVA with a Dunnett’s post test was used for multiple comparisons and the Mann–Whitney *U* test for pairwise comparisons. IC_50_ values were calculated with nonlinear 3-parametric regression.

All experiments with SH-SY5Y, *E. coli* and *B. subtilis* cell cultures were performed in 3 independent repetitions. The error in the luminescence measurements was calculated using the standard deviation equation based on 3 repeats, its values on the graphs are shown with error bars. EC_50_ concentrations were determined using Combenefit software [45,46].

### 3.13. SOS Chromotest

The SOS chromotest was carried out according to the standard procedure [47], with recommended additions [48]. The *E. coli* PQ37 strain was provided by P. Quillardet (Paris, France). The test compounds were dissolved in dimethyl sulfoxide (DMSO) (also used as a negative control) and added to the bacterial medium in the required concentrations; the concentration of the injected solutions was 1% by volume. The standard genotoxic agent 4-nitroquinoline-1-oxide (NQO) was chosen as the positive control.

### 3.14. Evaluation of Antioxidant Activity

#### 3.14.1. DPPH Radical Scavenging Activity

The free radical scavenging activity was evaluated using the stable radical 2,2-diphenyl-1-picrylhydrazyl (DPPH), according to the method described by Brand-Williams with a slight modification. Solutions of the compounds in MeOH were studied at a concentration of 0.2 mM. The stock DPPH solution contained 0.2 mM of radical in MeOH. An amount of 0.1 mL of the test compound solution was added to 0.1 mL of DPPH solution (0.2 mM) in each cell so that the initial DPPH concentration in the cells was 0.1 mM. The microplate was placed in a spectrophotometer, and the decrease in the absorbance values of the DPPH solution for 40 min at 20 °C was measured at λ_max_ = 517 nm. The results were expressed as scavenging activity, calculated as follows: scavenging activity, % = [(A_0_ − A_1_)/A_0_] × 100, where A_0_ is the optical density of the control solution DPPH, and A_1_ is the optical density of the reaction mixture solution.

#### 3.14.2. Cupric Reducing Antioxidant Capacity (CUPRAC Assay)

Neocuproine (2,9-dimethyl-1,10-phenanthroline) and Trolox were used with no further purification. The method proposed by Apak et al. was used with slight modifications. For these measurements, 0.05 mL of CuCl_2_ solution (0.01 M), 0.05 mL of MeOH neocuproine solution (7.5 mM) and 0.05 mL of ammonium acetate buffer solution (1 M) were added to a test tube, followed by mixing with the 0.05 mL tested compounds (0.5 mM). The mixtures were kept at room temperature for 30 min. The absorbance was measured at 450 nm against a reagent blank. The results were presented in Trolox equivalents (Trolox equivalent antioxidant capacity, TEAC) obtained using absorbance data, and the linear calibration curve was plotted as the absorbance vs. Trolox concentration.

#### 3.14.3. Ferric-Reducing Antioxidant Power (FRAP Assay)

The sample solution (0.1 mL) was mixed with 0.1 mL phosphate buffer (0.2 M, pH 6.6) and 0.1 mL potassium ferricyanide (1%). For each test compound, different concentrations of EtOH were used (5, 10, 25, 50, 100, 200 and 250 μM). The resulting mixture was incubated at 50 °C for 20 min. After the incubation period, 0.1 mL trichloroacetic acid (10%), 0.5 mL deionized water and 0.1 mL ferric chloride (0.1%) were added to the mixture. The sample absorbance was read at 700 nm with a 96-cell microplate spectrophotometer. The reduction of Fe (III) to Fe (II) could be expressed as % inhibition or the equivalent of a standard compound (e.g., Trolox).

#### 3.14.4. Ferrous Ions (Fe^2+^) Chelating Activity (FIC)

The chelation of ferrous ions by compounds was estimated by the method of Dinis et al. Briefly, 10 μL of 2 mM FeCl_2_ was added to 20 μL of the investigated compound (5 mM) and 150 μL of EtOH. The reaction was initiated by the addition of 0.04 mL of 5 mM ferrozine solution. The mixture was left to stand at 35 °C for 10 min. The absorbance of the solution was thereafter measured at 562 nm. The percentage inhibition of ferrozine–Fe^2+^ complex formation was calculated as [(A_0_ − A_s_)/A_s_] × 100, where A_0_ is the absorbance of the control, and A_s_ is the absorbance of the compound/standard. Na_2_EDTA was used as the positive control.

#### 3.14.5. Nitric Oxide Radical Scavenging Assay

Sodium nitroprusside (SNP) dissolved in aqueous solution at physiological pH 7.2 can generate nitric oxide whose level can be measured by the Griess reaction. Sample solution (50 µL, 0.5 mM) was mixed with SNP (50 µL, 5 mM) in phosphate buffer at pH 7.4 (0.2 M), followed by an incubation at room temperature for 150 min. Similarly, a blank was prepared by adding the sample solution in phosphate buffer without SNP. Griess reagent (solutions of 0.33% sulfanilic acid and 0.1% N-(1-naphthyl) ethylenediamine (NED) (C_12_H_14_N_2_) in 20% glacial acetic acid; 1:1) (100 µL) was added to the incubated sample and allowed to stand for 30 min. Absorbance values of the blank and samples were measured at 548 nm. The absorbance of the blank was then subtracted from that of the sample. Results were expressed as %inhibition. Inhibition I (%) = [(A_0_ − A_1_)/A_0_] × 100, where A_1_ is the absorbance in the presence of the testing compound, and A_0_ is the absorbance of the blank solution. All experiments were performed three times.

#### 3.14.6. Inhibition of Superoxide Radical Anion Formation by Xanthine Oxidase (NBT Assay)

EDTA, xanthine, bovine serum albumin, nitro blue tetrazolium and xanthine oxidase (25 MU) were purchased from Sigma Aldrich (Burlington, MA, USA). The superoxide anions were generated enzymatically by the xanthine oxidase system. The reaction mixture consisted of 2.70 mL of 40 mM sodium carbonate buffer containing 0.1 mM EDTA (pH 10.0), 0.06 mL of 10 mM xanthine, 0.03 mL of 0.5% bovine serum albumin, 0.03 mL of 2.5 mM nitro blue tetrazolium and 0.06 mL of the sample solution in DMSO at the concentration of 5 mM. An amount of 0.12 mL of xanthine oxidase (0.04 units) was added to the mixture at 25 °C, and the absorbance at 560 nm (by formation of blue formazan) was measured by a microplate spectrophotometer for 60 s. A control experiment was carried out by replacing the sample solution with the same amount of DMSO.

Inhibition I (%) = [(1 − A_i_/A_0_) × 100%], where A_i_ is the absorbance in the presence of the testing compound, and A_0_ is the absorbance of the blank solution. All experiments were performed three times.

#### 3.14.7. SOD-Protective Activity and Pro-/Antioxidant Activity

SOD-protective activity of the biopreparation was the ability to utilize the superoxide anion radical O_2_^–•^, as determined by the method of Sirota. A cytosolic fraction of the Russian sturgeon liver homogenate was used as the source of SOD. First, the sturgeon liver was washed with cold 0.2 M Tris (tris (hydroxymethyl) aminomethane) buffer (pH 7.8) to remove any traces of blood. All of the procedures were performed at a temperature of 0–4 °C. Next, a homogenate was obtained using a Potter homogenizer (Thomas Scientific, Swedesboro, NJ, USA) in 0.2 M Tris buffer at a ratio of 1:10. The homogenate was then centrifuged for 10 min at 1000× *g* to remove partially destroyed cells and nuclei. The resulting supernatant contained the enzymes of the cytosolic fraction of the liver homogenate, including the SOD. Here, 10 µL of the biopreparation was added to a cuvette with 200 µL of bicarbonate buffer (pH 10.65), 10 µL of the tested compound (initial concentrations of these compounds 25 µM) and 10 µL of 0.1% adrenaline solution and was thoroughly and quickly mixed. The rate of adrenaline oxidation without and in the presence of the biopreparation was evaluated by the change in optical density, measured at 347 nm for 3 min. The decrease in the rate of the process in the presence of the biopreparation was used to characterize the SOD-protective activity.

## 4. Conclusions

In conclusion, selenoureas containing adamantane and aromatic groups yielded a series of sEH inhibitors with relatively high potency. It was also shown that cytotoxic concentrations were 17–88 times higher than the effective inhibitory concentrations against sEH. The inhibition of proinflammatory macrophage activation by compounds **4a** and **5b** may serve as an additional benefit for the treatment of inflammatory conditions. These data show that a selenourea pharmacophore can be used to generate potent inhibitors of the sEH. These data open the door to the preparation of new groups of sEHI to reduce endoplasmic stress and thus resolve inflammation and restore cell homeostasis in disease stress.

## Data Availability

The data presented in this study are available in Appendix A.

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
