# Peer review of "Anti-Inflammatory Activity of Soluble Epoxide Hydrolase Inhibitors Based on Selenoureas Bearing an Adamantane Moiety"

_ijms, 2022, doi:10.3390/ijms231810710_

Round 1
Reviewer 1 Report
The article “Anti-inflammatory activity of soluble epoxide hydrolase inhibitors based on selenoureas bearing an adamantane moiety” realized by Burmistrov and col. is part of a series of research on the discovery of new inhibitors of the soluble epoxide hydrolase (sEH) based on selenourea moiety. In this study, the authors describe the synthesis of 14 novels 1-[(adamantan-1-yl)alkyl]-3-phenyl- (4a-g) and benzylselenoureas (5a-g). The new compounds were tested for their biological activity. The authors found that compound 4a has a drug profile.
The research is well done, but requires some additions or corrections:
· I recommend the authors write mL and µL instead of ml or µl.
· Because the authors stated that the structure of the obtained compounds was confirmed by 1H and 13C NMR spectroscopy, GC-MS, and elemental analysis, these spectral data should be presented. I recommend authors to discuss the structural characteristics of the compounds that were proven by spectral analysis, and the mass and NMR spectra should be added as Supplementary Materials.
· It would have been nice if the authors had done the FTIR spectra for a better characterization of the obtained compounds.
· I recommend to review lines 285-290 and correct the sentences.
· The presentation of the references in the text of the article must be changed. In the text, reference numbers should be placed in square brackets [ ], and placed before the punctuation; for example [1], [1–3] or [1,2].
I consider that the article is interesting and that it has elements of originality, and if it will be completed and corrected, it can be published.
Author Response
The article “Anti-inflammatory activity of soluble epoxide hydrolase inhibitors based on selenoureas bearing an adamantane moiety” realized by Burmistrov and col. is part of a series of research on the discovery of new inhibitors of the soluble epoxide hydrolase (sEH) based on selenourea moiety. In this study, the authors describe the synthesis of 14 novels 1-[(adamantan-1-yl)alkyl]-3-phenyl- (4a-g) and benzylselenoureas (5a-g). The new compounds were tested for their biological activity. The authors found that compound 4a has a drug profile.
The research is well done, but requires some additions or corrections:
- I recommend the authors write mL and µL instead of ml or µl.
Corrected
- Because the authors stated that the structure of the obtained compounds was confirmed by 1H and 13C NMR spectroscopy, GC-MS, and elemental analysis, these spectral data should be presented. I recommend authors to discuss the structural characteristics of the compounds that were proven by spectral analysis, and the mass and NMR spectra should be added as Supplementary Materials.
NMR and mass Spectra were added as Supplementary Materials
- It would have been nice if the authors had done the FTIR spectra for a better characterization of the obtained compounds.
It does not seem possible for us to obtain FTIR spectra within the time limits specified by the editors for manuscript revision. We presume that 1H and 13C NMR spectroscopy, GC-MS, and elemental analysis are enough to fully confirm the structure of the reported compounds. We will try to add FTIR spectra to our further manuscripts.
- I recommend to review lines 285-290 and correct the sentences.
Sentences corrected
- The presentation of the references in the text of the article must be changed. In the text, reference numbers should be placed in square brackets [ ], and placed before the punctuation; for example [1], [1–3] or [1,2].
Corrected
I consider that the article is interesting and that it has elements of originality, and if it will be completed and corrected, it can be published.
Thank you for the review.

Reviewer 2 Report
The manuscript presents a series of epoxide hydrolase inhibitors based on an adamant moiety. The novelty of this work is the introduction of adamantane moiety in the synthesis of this specific inhibitor.
Abstract: ok
Introduction:
Please expand paragraphs 46-48 ( what diseases exactly, mode of action). What is the prototype, core, of the active compound, pharmacophore? Please elaborate ( a few phrases). Please expand the phrase regarding EC5026.
Results and discussions: As a suggestion: the authors should carry on some computational studies regarding compounds and interaction with their targets. Compounds 4a, 5b, and 5d should be discussed in interaction with soluble epoxide hydrolase.
Materials and methods: Some pictures with the culture cells’ growth inhibition would help.
Conclusions: ok
Author Response
The manuscript presents a series of epoxide hydrolase inhibitors based on an adamant moiety. The novelty of this work is the introduction of adamantane moiety in the synthesis of this specific inhibitor.
Abstract: ok
Introduction:
Please expand paragraphs 46-48 ( what diseases exactly, mode of action). What is the prototype, core, of the active compound, pharmacophore? Please elaborate ( a few phrases). Please expand the phrase regarding EC5026.
Lines 46-48 expanded with exact diseases. Pharmacophore and typical structure of sEH inhibitors added at lines 51-52. Phrase regarding EC5026 expanded to mention that it is a new drug candidate intended to treat neuropathic pain.
Results and discussions: As a suggestion: the authors should carry on some computational studies regarding compounds and interaction with their targets. Compounds 4a, 5b, and 5d should be discussed in interaction with soluble epoxide hydrolase.
Discussion of compounds 4a, 5b, and 5d added in lines 119-122. Extensive computational studies of ureas and thioureas interactions with sEH were reported in our previous publications. In this case we suggest that selenoureas would not have any specific interactions different from ureas. Moreover, most of computational software could not emulate selenium with high confidence. However, we will try to provide such data in our future publications.
Materials and methods: Some pictures with the culture cells’ growth inhibition would help.
Cell studies were made to prove the absence of toxic effects. For all of the compound no cell death observed within the IC50 range of concentrations (below 1 µM). For most of the compounds no cell death observed at higher concentrations. In this case, the pictures will not be clear, since cell death was not observed.
Conclusions: ok
Thank you for the review.

Reviewer 3 Report
The manuscript is devoted to the study of the anti-inflammatory activity of selenoureas based on the inhibition of soluble epoxide hydrolase. The data presented in the article may be of interest to specialists in the field of medicinal chemistry, but the chemical part of the article does not contain new data. All target compounds were described in a previously published article (ref. 13), but the authors forget to cite this article in the relevant parts of the manuscript, in addition, the information about financial support of the synthetic part of this work by the Russian Fund for Basic Research (grant number 20-03-00298) seems inappropriate. Therefore, in my opinion, manuscript should be revised with a focus on the biological activity of these compounds or the synthetic part should be expanded. Also some comments are given below.
Abstract: A series of inhibitors of the soluble epoxide hydrolase (sEH) based on selenourea moiety and containing adamantane and aromatic lipophilic groups was developed. Delete this sentence since this series of inhibitors was developed in a previously published article (ref. 13).
Lines 63-65: NMR studies of ureas, thioureas and selenoureas revealed that properties of selenium-containing compounds are very close to its sulfur analogs.13 Please clarify what properties you mean other than the chemical shifts of these compounds.
All schemes: All compounds in the schemes should be numbered
Scheme 1: Correct the reaction conditions on the diagram: NaOH (50%, водн.) should be aq.
Lines 79-86: The data of this paragraph repeat the content of the previously published article (ref. 13). You should either cite the previous article, or delete the paragraph.
Line 328, 3.1. Synthetic procedures: cite ref. 13
Author Response
The manuscript is devoted to the study of the anti-inflammatory activity of selenoureas based on the inhibition of soluble epoxide hydrolase. The data presented in the article may be of interest to specialists in the field of medicinal chemistry, but the chemical part of the article does not contain new data. All target compounds were described in a previously published article (ref. 13), but the authors forget to cite this article in the relevant parts of the manuscript, in addition, the information about financial support of the synthetic part of this work by the Russian Fund for Basic Research (grant number 20-03-00298) seems inappropriate. Therefore, in my opinion, manuscript should be revised with a focus on the biological activity of these compounds or the synthetic part should be expanded. Also some comments are given below.
Thank you for the review. It should be noted that in order to perform all biological tests of the compounds, it was necessary to synthesize larger quantities, during which the methods for their preparation and isolation were improved, which is not possible without the use of Russian Fund for Basic Research grant funds.
Abstract: A series of inhibitors of the soluble epoxide hydrolase (sEH) based on selenourea moiety and containing adamantane and aromatic lipophilic groups was developed. Delete this sentence since this series of inhibitors was developed in a previously published article (ref. 13).
Sentence deleted
Lines 63-65: NMR studies of ureas, thioureas and selenoureas revealed that properties of selenium-containing compounds are very close to its sulfur analogs.13 Please clarify what properties you mean other than the chemical shifts of these compounds.
It is only chemical shifts. Sentence corrected.
All schemes: All compounds in the schemes should be numbered
Corrected
Scheme 1: Correct the reaction conditions on the diagram: NaOH (50%, водн.) should be aq.
Corrected
Lines 79-86: The data of this paragraph repeat the content of the previously published article (ref. 13). You should either cite the previous article, or delete the paragraph.
Citation added.
Line 328, 3.1. Synthetic procedures: cite ref. 13
Citation added.

Reviewer 4 Report
The authors have synthesized a series of selenoureas as soluble epoxide hydrolase inhibitors as anti-inflammatory agents. A structure-activity relationship study was presented. The series was tested for their anti-inflammatory activity using nitric oxide synthesis assay. The safety of the synthesized compounds was assessed over several cell lines along with other assays. The manuscript represents a continuation of a study initiated by the authors for ureas, thioureas as sEH inhibitors. While being well organized, the manuscript has several major flaws that must be addressed before being considered for publication. Please refer to the list of comments below.
1. Detailed experimental procedures, chemical yields and characterization of the compounds were not included in the manuscript. This is mandatory. 2. The schemes are missing the reaction conditions. 3. The manuscript should be written in English only according to the journal requirements. 4. A figure showing some of the sEH inhibitors, such as EC5026 mentioned in the text, would be helpful to the reader. 5. Another figure showing the basis of the design of the new inhibitors is recommended. 6. Line 84: “The purity was 99+% (GC-MS).” Please provide purity data as a supporting information along with the characterization raw data. 7. The results and discussion section should be subcategorized into a chemistry section and a Structure-Activity Relationship study section (from line 100), … etc for the convenience of the reader. 8. The header of table 1 need to be corrected. 9. It is recommended to add reference standard to table 1. 10. I am curious to see if COX inhibition has anycontribution to the observed anti-inflammatory activity of the new compounds. 11. Line 175: “indicates the absence of a toxic effect on bacterial cells.” As the compounds are intended as anti-inflammatory, the importance of the quoted sentence to the study needs further explanation. 12. It is too early to claim that a series of compounds have anti-inflammatory activity with only nitric oxide test. Other presented assays claim the safety of the new series. Further assays that include other inflammatory mediators are needed. 13. The relation between the anti-inflammatory and the anti-radical activity needs to be discussed in the manuscript otherwise the reader could think that those assays are irrelevant to the study. 14. The ability of compound 4a to induce oxidative stress could be marketed as anti-proliferative activity, which is irrelevant to the study. Please explain. 15. Attention is needed to the language and sentences of the manuscript as well as the formatting.
Author Response
The authors have synthesized a series of selenoureas as soluble epoxide hydrolase inhibitors as anti-inflammatory agents. A structure-activity relationship study was presented. The series was tested for their anti-inflammatory activity using nitric oxide synthesis assay. The safety of the synthesized compounds was assessed over several cell lines along with other assays. The manuscript represents a continuation of a study initiated by the authors for ureas, thioureas as sEH inhibitors. While being well organized, the manuscript has several major flaws that must be addressed before being considered for publication. Please refer to the list of comments below.
- Detailed experimental procedures, chemical yields and characterization of the compounds were not included in the manuscript. This is mandatory.
Detailed experimental procedures and characterization added to section 3.1
- The schemes are missing the reaction conditions.
Reaction conditions added to Scheme 2 and 3
- The manuscript should be written in English only according to the journal requirements.
Corrected
- A figure showing some of the sEH inhibitors, such as EC5026 mentioned in the text, would be helpful to the reader.
Figure showing some of the known soluble epoxide hydrolase inhibitors was added.
- Another figure showing the basis of the design of the new inhibitors is recommended.
There are too many ways of sEHI design and we think that such figure will overload the introduction. Especially since the modification of inhibitors is not the aim of this work.
- Line 84: “The purity was 99+% (GC-MS).” Please provide purity data as a supporting information along with the characterization raw data.
Purity data added as a supporting information along with the NMR and mass spectra
- The results and discussion section should be subcategorized into a chemistry section and a Structure-Activity Relationship study section (from line 100), … etc for the convenience of the reader.
Subsections added to the results and discussion section
- The header of table 1 need to be corrected.
Corrected
- It is recommended to add reference standard to table 1.
Reference standard added to table 1
- I am curious to see if COX inhibition has any contribution to the observed anti-inflammatory activity of the new compounds.
We will try to provide such data in our future publications.
- Line 175: “indicates the absence of a toxic effect on bacterial cells.” As the compounds are intended as anti-inflammatory, the importance of the quoted sentence to the study needs further explanation.
As soon as a new chemical, prominent anti-inflammatory compound shows cytotoxic effect, it should be investigated comprehensively. To do that we tested different types of cells, including bacteria. Bacterial tests showed the absence of specific and general toxicity in concentrations below 50 µM, which was reported.
- It is too early to claim that a series of compounds have anti-inflammatory activity with only nitric oxide test. Other presented assays claim the safety of the new series. Further assays that include other inflammatory mediators are needed.
We will try to provide such data in our future publications.
- The relation between the anti-inflammatory and the anti-radical activity needs to be discussed in the manuscript otherwise the reader could think that those assays are irrelevant to the study.
Many organoselenium compounds exhibit antioxidant activity which correlates with their ability to scavenge reactive oxygen species (ROS). Moreover, it was recently shown that selenoureas of the similar structure possess moderate antioxidant activity. In this case such assays in our opinion are relevant to the current study. This additional activity enhances the medicinal potential of the compounds. Linking sentences added to the manuscript (lines 238-240).
- The ability of compound 4a to induce oxidative stress could be marketed as anti-proliferative activity, which is irrelevant to the study. Please explain.
It was investigation of 4a toxicity during which its ability to induce oxidative stress in bacterial cells was discovered. It obviously could not be marketed as anti-proliferative activity since 4a needs to be at very high concentration (one to several orders higher than IC50 (table 1) or lethal for eukaryotic cells concentration) to cause oxidative stress. This difference in concentrations is mentioned in manuscript (lines 199-202).
- Attention is needed to the language and sentences of the manuscript as well as the formatting.
The manuscript has been revised again

Reviewer 5 Report
The present manuscript entitled "Anti-inflammatory activity of soluble epoxide hydrolase inhibitors based on selenoureas bearing an adamantane moiety" reports the synthesis of a series of inhibitors of the soluble epoxide hydrolase based on selenourea moiety and containing adamantane and aromatic lipophilic groups. In general, the manuscript presents a diverse set of chemical and biological data related to the medicinal chemistry approach. Despite the satisfactory description of the structure-biological activity relationship, It is necessary to improve the relationship between the other parts of the tests performed. In addition, there is some information missing from the manuscript. Therefore, it is not possible to accept it for publication in this journal. Please note the comments below:
- The manuscript does not have the ethics committee's approval for the use of animals in the laboratory. Authors are required to enter the approval registration number;
- The manuscript does not describe the chemical data of structural characterization of the synthesized compounds. These data are essential to prove the proper synthesis of the compounds. Unpublished compounds must have their NMR spectra in a supplementary file.
minor revisions
- Please improve the explanation of the pharmacological importance of the inhibitors of the soluble epoxide hydrolase in the first paragraph of the Introduction. The relationship between metabolism and clinical application is not understandable.
- Correct the sentence in line 160 "4g in in different concentrations";
- In lines 190-191 replace the term "The lack of toxicity" with "the absence of toxic effects";
- Please inform the manuscript, experimental part, the age (weeks) of the mice used;
- In Conclusions authors wrote "These data show that a selenourea pharmacophore can be used to generate potent inhibitors". Please clarify this sentence. Molecular modeling was performed in this study to suggest the pharmacophore?
Author Response
The present manuscript entitled "Anti-inflammatory activity of soluble epoxide hydrolase inhibitors based on selenoureas bearing an adamantane moiety" reports the synthesis of a series of inhibitors of the soluble epoxide hydrolase based on selenourea moiety and containing adamantane and aromatic lipophilic groups. In general, the manuscript presents a diverse set of chemical and biological data related to the medicinal chemistry approach. Despite the satisfactory description of the structure-biological activity relationship, It is necessary to improve the relationship between the other parts of the tests performed. In addition, there is some information missing from the manuscript. Therefore, it is not possible to accept it for publication in this journal. Please note the comments below:
Thank you for the review. Corrections were made in lines 120-122, 136-143, 184-185, 197-198, and 222-223 for better relationship between parts of the manuscript.
- The manuscript does not have the ethics committee's approval for the use of animals in the laboratory. Authors are required to enter the approval registration number;
Registration number added to paragraph 3.2
- The manuscript does not describe the chemical data of structural characterization of the synthesized compounds. These data are essential to prove the proper synthesis of the compounds. Unpublished compounds must have their NMR spectra in a supplementary file.
NMR and mass Spectra were added as Supplementary Materials
minor revisions
- Please improve the explanation of the pharmacological importance of the inhibitors of the soluble epoxide hydrolase in the first paragraph of the Introduction. The relationship between metabolism and clinical application is not understandable.
Paragraph improved
- Correct the sentence in line 160 "4g in in different concentrations";
Corrected
- In lines 190-191 replace the term "The lack of toxicity" with "the absence of toxic effects";
Corrected
- Please inform the manuscript, experimental part, the age (weeks) of the mice used;
Age of the mice used (4-5 weeks) added in paragraph 3.2
- In Conclusions authors wrote "These data show that a selenourea pharmacophore can be used to generate potent inhibitors". Please clarify this sentence. Molecular modeling was performed in this study to suggest the pharmacophore?
This assumption based on the biological data presented in the manuscript

Round 2
Reviewer 1 Report
Reading the article, I believe that the authors have increased the scientific quality.
I did not find in the Results and Discussions section comment on the most important NMR spectral characteristics of the studied compounds, as I suggested. I still recommend this, because it is very important for the reader to understand how the spectral interpretation was made.
I understood that the IR spectra were not taken and I agree that there was insufficient time. The authors should still carry out this spectral study.
I believe that if the authors complete the comments regarding the spectral characteristics, the article meets the criteria for publication.
Author Response
Reading the article, I believe that the authors have increased the scientific quality.
I did not find in the Results and Discussions section comment on the most important NMR spectral characteristics of the studied compounds, as I suggested. I still recommend this, because it is very important for the reader to understand how the spectral interpretation was made.
NMR spectra comments added to Results and Discussions section (lines 102-110).
I understood that the IR spectra were not taken and I agree that there was insufficient time. The authors should still carry out this spectral study.
We managed to carry out IR study of some of the compounds. C–Se stretching vibrations added to experimental section and original spectra to the supplementary.
I believe that if the authors complete the comments regarding the spectral characteristics, the article meets the criteria for publication.
Thank you for the review.

Reviewer 3 Report
The manuscript has been improved and can be published in present form. Also I recommend to add numbers of corresponding amines 1',2'(line 82)
Author Response
Reviewer 3
The manuscript has been improved and can be published in present form. Also I recommend to add numbers of corresponding amines 1',2'(line 82)
Corrected
Thank you for the review.

Reviewer 4 Report
The reviewer would like to thank the authors for taking the effort to improve their manuscript.
However, the authors failed to respond in a satisfactory way to critical points related to the relevance of the biological assays provided and consequently the significance of the content to the readers of IJMS readers (comments 10–14) as below.
1. The compounds reported herein are not novel and their synthesis and characterization were previously reported. The authors did not indicate that in their original submission. As the compounds were previously reported, there is no need to show the schemes or discuss the synthesis further and a citation of the relevant article would suffice.
2. As the compounds are introduced as anti-inflammatory agents, sufficient evidence shall be provided rather than general biological assays that proves the safety of the compounds and are not related to the claimed anti-inflammatory activity.
3. “For compounds with evident anti-inflammatory activity (4a, and 5b) and one out group compound (4g)”. the anti-inflammatory activity of 4a is not evident using one general anti-inflammatory assay. 5a is 17-fols less potent than 4a!
4. “Thus, the investigated compounds don’t show general or specific”. please do not use contraction forms in the text as it is not academic.
Therefore, I recommend against publishing this article in IJMS and I advise the authors to provide further assays to prove the anti-inflammatory activity of this series of organoselenium compounds and resubmit their manuscript.
Author Response
The reviewer would like to thank the authors for taking the effort to improve their manuscript.
However, the authors failed to respond in a satisfactory way to critical points related to the relevance of the biological assays provided and consequently the significance of the content to the readers of IJMS readers (comments 10–14) as below.
- The compounds reported herein are not novel and their synthesis and characterization were previously reported. The authors did not indicate that in their original submission. As the compounds were previously reported, there is no need to show the schemes or discuss the synthesis further and a citation of the relevant article would suffice.
Another reviewer asked us to comment NMR spectral characteristics of the studied compounds. Taking into account that manuscript was submitted to the special issue more focused on selenium containing compounds rather than bioactivity itself we believe that methods for the preparation of the studied compounds will not disturb readers.
- As the compounds are introduced as anti-inflammatory agents, sufficient evidence shall be provided rather than general biological assays that proves the safety of the compounds and are not related to the claimed anti-inflammatory activity.
In our work, we did not set the goal of a comprehensive study of the anti-inflammatory activity of these substances, but only an assessment of the potential possibility of their use as anti-inflammatory drugs. A comprehensive study of their anti-inflammatory activity is planned for the next stage. Corresponding corrections in the text were made (lines 150-152 and 158-160).
- “For compounds with evident anti-inflammatory activity (4a, and 5b) and one out group compound (4g)”. the anti-inflammatory activity of 4a is not evident using one general anti-inflammatory assay. 5a is 17-fols less potent than 4a!
The choice of 4a, 4g, and 5b for more detailed toxicological studies was made according to their sEH inhibitory activity and for NO synthesis inhibition. (lines 169-173).
- “Thus, the investigated compounds don’t show general or specific”. please do not use contraction forms in the text as it is not academic.
Thanks for the note, the wording has been corrected (lines 219-220).
Therefore, I recommend against publishing this article in IJMS and I advise the authors to provide further assays to prove the anti-inflammatory activity of this series of organoselenium compounds and resubmit their manuscript.

Reviewer 5 Report
The manuscript revised is ok.
Author Response
The manuscript revised is ok.
Thank you for the review.
